# The Dynamics of Gene Expression Unraveling the Immune Response of *Macrobrachium rosenbergii* Infected by *Aeromonas veronii*

**DOI:** 10.3390/genes14071383

**Published:** 2023-06-30

**Authors:** Xin Peng, Xuan Lan, Zhenxiao Zhong, Haihui Tu, Xinyi Yao, Qiongying Tang, Zhenglong Xia, Guoliang Yang, Shaokui Yi

**Affiliations:** 1Key Laboratory of Aquatic Animal Genetic Breeding and Nutrition, Chinese Academy of Fishery Sciences, Huzhou University, Huzhou 313000, China; pengxin0521@163.com (X.P.); lanxuan0570@163.com (X.L.); zhongzhenxiao1218@163.com (Z.Z.); tuhaihui09@163.com (H.T.); yaoxinyi2002@163.com (X.Y.); ygl0572@163.com (G.Y.); 2Zhejiang Provincial Key Laboratory of Aquatic Resources Conservation and Development, Huzhou University, Huzhou 313000, China; 3Jiangsu Shufeng Prawn Breeding Co., Ltd., Gaoyou 225654, China; zjhill@126.com

**Keywords:** transcriptome, differentially expressed genes, *Macrobrachium rosenbergii*, *Aeromonas veronii*

## Abstract

To further investigate the immune response of *Macrobrachium rosenbergii* against *Aeromonas veronii*, comparative transcriptomic analyses of the *M. rosenbergii* hepatopancreas were conducted on challenge and control groups at 6, 12, and 24 h post-infection (hpi), independently. A total of 51,707 high-quality unigenes were collected from the RNA-seq data, and 8060 differentially expressed genes (DEGs) were discovered through paired comparisons. Among the three comparison groups, a KEGG pathway enrichment analysis showed that 173 immune-related DEGs were considerably clustered into 28 immune-related pathways, including the lysosome, the phagosome, etc. Moreover, the expression levels of the four key immune-related genes (*TOLL*, *PAK1*, *GSK3β*, and *IKKα*) were evaluated at various stages following post-infection in the hepatopancreas, hemolymph, and gills. Both *PAK1* and *GSK3β* genes were highly up-regulated in all three tissues at 6 hpi with *A. veronii*; *TOLL* was up-regulated in the hepatopancreas and hemolymph but down-regulated in the gill at 6 hpi, and *IKKα* was up-regulated in hemolymph and gill, but down-regulated in the hepatopancreas at 6 hpi. These findings lay the groundwork for understanding the immune mechanism of *M. rosenbergii* after contracting *A. veronii*.

## 1. Introduction

The production of *M. rosenbergii* has reached an impressive 290,708 tons globally in 2021 [1]. China is a prominent producer, having produced a total of 171,263 tons in 2021 as reported by the “China Fisheries Yearbook,” with Guangdong, Jiangsu, and Zhejiang provinces contributing over 92% of the national total. However, the diversity and complexity of diseases affecting *M. rosenbergii* have resulted in substantial losses for farmers in recent years. These pathogenic microorganisms that invade *M. rosenbergii* include *M. rosenbergii* nodavirus (MrNV) and extra-small virus (XSV) [2], infectious precocity virus (IPV) [3], decapod iridescent virus 1 (DIV1) [4], *Citrobacter freundii* [5], etc.

*A. veronii* is a conditioned pathogen, widely distributed in fresh water, seawater, and soil [6], and was classified in the genus *Aeromonas* in 1978 [7]. As a Gram-negative bacterium, it has caused the mortality of various aquatic animals, such as Nile tilapia (*Oreochromis niloticus*) [8], yellow catfish (*Pelteobagrus fulvidraco*) [9], *Odontobutis potamophila* [10], etc. Our previous research discovered a dominant strain named WSQ-1, which has caused the mass mortality of adult *M. rosenbergii* in certain farms in Gaoyou city of Jiangsu Province. This strain was isolated from the hepatopancreas of dying *M. rosenbergii* and identified as *A. veronii*. The challenge test, amplification of virulence genes, and histopathological examination of the hepatopancreas all demonstrated that the strain was extremely pathogenic [11].

As an invertebrate, *M. rosenbergii* mainly relies on innate immunity to recognize and resist invading pathogens [12]. Among these tissues, the hepatopancreas is closely related to nutrient metabolism regulation [13], and the hemolymph plays a crucial role in the host immune response, including the recognition and phagocytosis of pathogenic bacteria [14,15]. The gills are more prone to damage and physiological responses compared to other organs as they participate in the absorption of pathogens into the hemolymph to eliminate them [16,17].

In previous experiments, we explored the virulence and pathogenicity of *A. veronii* [11]. In this study, we further investigated the immune response of *M. rosenbergii* against *A. veronii.* We conducted a comparative transcriptomic analysis on the hepatopancreas of *M. rosenbergii* infected with *A. veronii* at different time points (6, 12, and 24 post-injection). Additionally, four immune-related genes were identified from the transcriptomic data, and their expression levels were analyzed in the hepatopancreas, hemolymph, and gills at different post-infection time points. These findings may provide insights into the host response to *A. veronii* infection and can be used as a guide for *A. veronii* prevention and treatment.

## 2. Materials and Methods

### 2.1. Experimental Prawns and A. veronii Preparation

The healthy prawns, which had an average body weight of 3.11 ± 0.49 g, were used in the challenge experiment. Before being subjected to experimental manipulation, all prawns were acclimatized for one week in the recirculating water barrel system of Yuya Technology (Huzhou) Co. LTD (Zhejiang, China). Since the suitable growth temperature range for *M. rosenbergii* is 24–32 °C, the water temperature was maintained at 28 ± 2 °C during the challenge experiment. The dissolved oxygen concentration was above 5 mg/L and the prawns were fed twice daily but were not fed during the injection trial. Then, 180 prawns were split into two groups (*n* = 90) (the challenge groups and the control groups) at random, and each group was further separated into three parallel groups so that there were 30 prawns in each 60-L plastic container. Random inspections of the prawns by plate culture test were conducted before the injection trial to ensure that they were free of any bacterial infections.

In accordance with previous tests’ median lethal dosage (LD50) of 72 h, the strain WSQ-1 (*A. veronii*) was first inoculated in the tryptic soybean peptone liquid medium (TSB), then placed in a 28 °C shaker with shaking at 180 rpm for 18 h. Lastly, the turbidity was adjusted to 2.50 × 10^6^ CFU/mL.

### 2.2. Sample Collection

In the experiment, the challenge groups received an intramuscular injection of 50 μL of *A. veronii*, whereas the control groups were injected with sterilized PBS in the same amount. After 6, 12, and 24 hpi, prawns from the two groups were randomly selected, and the hepatopancreas and gills were temporarily stored in liquid nitrogen and then transferred to the ultra-low temperature (−80 °C) refrigerator. Every three of the same tissues were mixed into one sample at different time points, with three biological replicates. Significantly, the treatment of the hemocytes was particularly important. Briefly, the fresh hemolymp and TRIzol were first added to the cryotube in a 1:3 ratio to lyse the blood cells, and the mixture was then shocked for 1–2 min until the floccule was completely cracked. Finally, the mixture was incubated at room temperature for 5 min to completely decompose the ribosome and then stored at −80 °C.

### 2.3. Total RNA Extraction and Illumina Sequencing

The TRIzol reagent (Dalian, China) was used to extract the total RNA from 18 hepatopancreas samples at 6, 12, and 24 hpi in accordance with the manufacturer’s instructions. The challenge groups were named AV6_1, AV6_2, AV6_3, AV12_1, AV12_2, AV12_3, AV24_1, AV24_2, AV24_3, whereas the control groups were marked as C6_1, C6_2, C6_3, C12_1, C12_2, C12_3, C24_1, C24_2, C24_3, separately. These libraries were obtained after the whole RNA was first screened and purified using the AMPure XP technology (Beckman Coulter, Beverly, CA, USA). Following these libraries’ initial Qubit 2.0 quantification, they were diluted to 1.5 ng/uL and identified using an Agilent 2100 bioanalyzer. These libraries’ quality was further ensured by using qRT-PCR to precisely measure their effective concentration. The libraries’ construction and sequencing were completed using the Illumina NovaSeq 6000.

### 2.4. De Novo Assembly, Annotation, and Classification

To ensure the accuracy and dependability of data analysis, some raw data, such as reads with adapters, low-quality, and N bases were filtered away to obtain clean reads. Next, GC content and Q30 were computed with clean data by fastp (version 0.19.7). Trinity was used to assemble the transcripts [18], and BUSCO software was used to evaluate the correctness and completeness of the transcripts [19]. Finally, all of the transcripts were classified and annotated using the following 7 databases: Nr, Nt, Pfam, COG, Swiss-Prot, KO, and GO.

### 2.5. DEGs, KEGG, and GO Enrichment Analysis

The DESeq2 package (version 1.20.1) [20] was used in this research to screen DEGs for significantly differential gene expression. In addition, we set three comparisons between challenge and control groups after 6, 12, and 24 hpi (i.e., AV6 vs. C6, AV12 vs. C12, and AV24 vs. C24); next, the immune-related DEGs were collected and analyzed for GO and KEGG enrichment using GOseq and KOBAS software packages, separately [21].

### 2.6. Validation of Immune-Related DEGs by RT-qPCR

Ten immune-related DEGs were chosen at random for quantitative real-time PCR (qRT-PCR) analysis in order to verify the credibility of the transcriptome sequencing data. Primer 5.0 software was used to design the primers (Table 1), which were then synthesized by the company. The cDNA of the hepatopancreas (AV6 vs. C6) used for qRT-PCR was synthesized by reverse transcription using cDNA Synthesis Supermix (TransGen Biotech, Beijing, China), while the 18S rRNA was used as the reference gene for the qRT-PCR. In addition, amplifications were performed in a 10 µL reaction system with 0.8 µL of cDNA, 3.4 µL of ddH_2_O, 5 µL of TB Green Premix Ex Taq II (2×), and 0.4 µL each PCR forward primer and reverse primer. The PCR procedure was pre-denaturation at 95 °C for 3 min, followed by 39 cycles of denaturation at 95 °C for 5 s and annealing at 59 °C for 30 s. When the temperature rose from 65 °C to 95 °C, the melting curve increased by 0.5 °C every 5 s. Each qRT-PCR assay was carried out in three biological replicates and three technique replicates. Ultimately, the 2^−ΔΔCT^ method was used to evaluate relative mRNA expression.

### 2.7. Expression Patterns of Four Key Immune-Related DEGs in Different Tissues

Based on KEGG enrichment pathway analysis results, *protein Toll* gene (*TOLL*) was enriched in “Toll and Imd signaling pathway”; *p21-activated kinase 1* (*PAK1*) was enriched in three immune-related pathways, such as “Chemokine signaling pathway”, etc.; *glycogen synthase kinase 3 β* (*GSK3β*) was enriched in six immune-related pathways, such as “mTOR signaling pathway”, etc.; and *inhibitor of nuclear factor kappa-B kinase subunit α* (*IKKα*) was enriched in 12 immune-related pathways, such as ”NOD-like receptor signaling pathway”, etc. (Table 2). In addition, *TOLL*, *PAK1*, *GSK3β* were up-regulated and *IKKα* was down-regulated at 6 hpi.

To further analyze the relative mRNA expression in different tissues and time points, these four immune-related DEGs were selected and anticipated to be engaged in the immunological response of the tissues. The collected samples at 6, 12, and 24 hpi were used to extract RNA and synthesize cDNA based on the previous methods, and all qRT-PCR reactions were performed in triplicate (three biological replicates and three technique replicates). Lastly, the 2^−ΔΔCT^ approach was used to calculate the relative mRNA expression.

### 2.8. Statistical Analysis

SPSS 25.0 software (IBM Corp., Armonk, NY, USA) was used to detect statistical significance for differences in gene expression levels between the challenge and control groups at three time points after infection in three tissues including hemolymph, hepatopancreas, and gills. Then, a chart was created with GraphPad Prism 9.0 software (GraphPad Software Inc., San Diego, CA, USA).

## 3. Results

### 3.1. Transcript Assembly, Gene Functional Annotation, GO, and KEGG Classification

There were a total of 22,248,733–24,166,064 raw reads in 18 libraries, and 21,489,762–23,722,890 clean reads were obtained after quality control (Appendix A). The Q30 percentages’ average values were 93.35%. Using the Trinity software, 93,091 transcripts were produced, with an average length of 1716 bp. Finally, 51,707 unigenes were obtained, with sizes ranging from 301 bp to 28,851 bp and the N50 length of 2808 bp (Figure 1A).

Seven public databases were used to annotate the functions of all unigenes: 17,353 (33.56%) in NR, 5130 (9.92%) in NT, 7354 (14.22%) in KEGG, 12,481 (24.13%) in Swiss-Prot, 16,723 (32.34%) in Pfam, 16,720 (32.33%) in GO, and 6487 (12.54%) in KOG, respectively (Appendix A). In addition, 2325 genes were coexisting functional genes annotated in the 5 databases, as demonstrated in the Venn diagram of Nr, Nt, Pfam, GO, and KOG annotations in Figure 1B.

KEGG is used to comprehend the sophisticated functions of biological systems. In this study, 861 unigenes were associated with “signal transduction”, 618 unigenes with “transport and catabolism”, and 366 unigenes with the “immune system” (Appendix A). Meanwhile, GO analysis is also important. 16,720 unigenes were classified as molecular function, cellular component, and biological process, with 12, 5, and 25 subcategories, respectively. Among the biological processes, 45 unigenes were associated with “antioxidant activity” (GO:0016209), 173 unigenes with “immune system process” (GO:0002376), and 2526 unigenes were associated with “response to stimulus” (GO:0050896), respectively (Appendix A).

### 3.2. Identification of DEGs Related to A. veronii Infection

In the current study, 8060 DEGs (Appendix A) were detected in three transcriptome comparisons between challenge and control groups (AV6 vs. C6, AV12 vs. C12, and AV24 vs. C24), among which 5208 DEGs (containing 2596 up-regulated and 2612 down-regulated genes) were involved at the early stage of the challenge (AV6 vs. C6). The DEGs reduced dramatically with the extension of the challenge time, with 2262 genes (1225 down-regulated and 1037 up-regulated) and 590 genes (325 down-regulated and 265 up-regulated) differentially expressed, respectively, at 12 and 24 hpi (Figure 2A). The Venn diagram showed that 36 DEGs coexisted in three comparable groups (Figure 2B).

### 3.3. KEGG Enrichment of the Immune-Related DEGs

At 6, 12, and 24 hpi, KEGG enrichment of immune-related DEGs in different immune-related pathways was analyzed between the challenge group and the control group. For the three transcriptome comparisons, 28 important immune-related pathways were chosen (Appendix A). As shown in Figure 3, the lysosome pathway was substantially enriched by all three comparisons, with the highest number of DEGs in the top-20 immune-related pathways. Except for Figure 3C, both comparison groups of AV6 vs. C6 (Figure 3A) and AV12 vs. C12 (Figure 3B) greatly enriched the phagosome pathway. A total of six up-regulated genes were detected at 6 hpi, including actin γ 1 *(F-actin*), C-type lectin domain family 4 member L/M (*DCSIGN*), tubulin β (*TUBB*), 1-phosphatidylinositol-3-phosphate 5-kinase (PIKFYVE), cation-dependent mannose-6-phosphate receptor (*M6PR*), and nitric-oxide synthase (*NOS1*) in the phagosome pathway (Figure 4A), and 4 up-regulated genes at 6 hpi were detected in the lysosome pathway, including lysosomal acid lipase (*LIPA*), lysophospholipase III (LYPLA3), lysosomal acid phosphatase (*ACP2*), and ceroid-lipofuscinosis neuronal protein 7 (*CLN7*) (Figure 4B). Other DEGs information was shown in Table 3 and other immune-related pathways, such as the MAPK signaling pathway-fly, the chemokine signaling pathway, antigen processing and presentation, etc., were also abundant in the DEGs.

Furthermore, the Venn diagram analysis showed that 173 immune-related DEGs were detected based on the aforementioned 28 immune-related pathways, and there were 138, 43, and 12 immune-related DEGs, respectively, at 6, 12, and 24 hpi compared to the control groups (Figure 5A). Furthermore, in the hierarchical cluster analysis, 173 immune-related DEGs in AV6 showed significant differences compared to AV12 and AV24, indicating that the majority of the 173 immune-related DEGs were relatively highly expressed at 6 hpi, then the number of DEGs was drastically reduced at 12 and 24 hpi; some genes’ expression increased at 12 hpi while being relatively low expressed at 6 and 24 hpi; a few genes were highly expressed at 24 hpi compared with 6 and 12 hpi (Figure 5B).

### 3.4. Validation of DEGs by qRT-PCR

To further confirm the gene expression from RNA-seq data, 10 immune-related DEGs, including *TOLL*, *GSK3β*, *IκB*, *ACTIN*, *TBK1*, *MALT*, *GRB2*, *PAK1*, *CD13*, and *JACK1/2*, were selected for qRT-PCR verification at random. The results showed that qRT-PCR had an identical expression tendency to the RNA-seq data, which confirmed the expression of immune-related DEGs in the RNA-seq data (Figure 6).

### 3.5. Temporal and Spatial Expression Levels of Four Key Immune-Related Genes in M. rosenbergii

To further explore the gene expression in hepatopancreas, hemolymph, and gills, four immune-related DEGs, including *TOLL*, *PAK1*, *GSK3β*, and *IKKα*, were screened from the KEGG pathways, in which these genes overlapped each other.

As seen in Figure 7A, the *TOLL* gene’s expression level in the hepatopancreas of the challenge groups at 6 hpi was significantly higher than that in the control group (*p* < 0.05), but then sharply decreased at 12 hpi, significantly lower than that in the control group (*p* < 0.05), and almost kept successively low levels at 24 hpi, indicating that this gene was remarkably up-regulated at 6 hpi, whereas it became down-regulated at 12 and 24 hpi. Similar to the hepatopancreas, the expression trend of *TOLL* in the hemolymph of the challenge group was significantly higher than that in the control group (*p* < 0.01) at 6 hpi, then slightly dropped at 12 hpi and remained similar at 24 hpi, still with a higher expression level than the control group, though the difference was not significant. Therefore, the expression of *TOLL* was up-regulated in the hemolymph after *A. veronii* infection. Unlike the hepatopancreas and hemolymph, in the gills, the expression of *TOLL* was down-regulated at 6 and 12 hpi, significantly lower than the control group at 6 hpi (*p* < 0.05), then slightly increased at 12 hpi, and became up-regulated at 24 hpi.

In the hepatopancreas, the *PAK1* gene expression in the challenge group at 6 hpi was higher than that in the control group, then slightly decreased at 12 hpi and dropped dramatically at 24 hpi, but both of them were significantly higher than the control group (*p* < 0.05), indicating that the *PAK1* gene was remarkably up-regulated at the three time points, despite the continuing decline at 12 and 24 hpi. A similar trend of *PAK1* gene expression also appeared in the hemolymph and gills, except that the expression level of the *PAK1* gene was slightly lower than that in the control group at 24 hpi in the gills and became down-regulated (Figure 7B).

In the hepatopancreas and gills, the expression trend of the *GSK3β* gene was similar to the *TOLL* gene expression in the hepatopancreas, which was significantly higher than that in the control group (*p* < 0.05), but precipitously declined at 12 and 24 hpi; both of the expression levels at the two points were slightly lower than those in the control group, indicating that the *GSK3β* gene was up-regulated at 6 hpi, whereas it became down-regulated at 12 and 24 hpi. Furthermore, the expression trend of the *GSK3β* gene in the hemolymph was similar to that of the *PAK1* gene expression in the hepatopancreas and hemolymph, which was higher than that in the control group at 6 hpi (*p* < 0.01) but sharply decreased at 12 hpi, in spite of the fact that the difference was not significant compared to the control group. Following that, it maintained a slight decline at 24 hpi, but was significantly higher than that in the control group (*p* < 0.01), revealing that this gene was up-regulated and expressed at all three points (Figure 7C).

The expression level of the *IKKα* gene in the hepatopancreas of the challenge groups at 6 hpi decreased slightly and maintained a sustained decline at 12 and 24 hpi, and all of them were lower than the control group, which revealed that this gene remained down-regulated throughout the challenge. Unlike the hepatopancreas, the expression level in the hemolymph was remarkably higher than the control group at 6 hpi (*p* < 0.01) but decreased sharply at 12 hpi and was lower than the control group. After that, it increased slightly at 24 hpi but was still lower compared to the control. These results revealed that the gene *IKKα* was up-regulated at 6 hpi and became down-regulated at 12 and 24 hpi. In the gill, the *IKKα* gene in the challenge groups was slightly higher at 6 hpi than the control group and was remarkably higher compared to the control group at 12 hpi (*p* < 0.01), but then sharply declined at 24 hpi and was lower than the control group, indicating that the *IKKα* gene was up-regulated at 6 and 12 hpi but became down-regulated at 24 hpi (Figure 7D).

## 4. Discussion

### 4.1. Phagosome and Lysosome Pathway Analysis

Phagocytosis plays an important role in the innate immunity of *M. rosenbergii* [15,22]. The phagocytosis process was the particle binding to the cell surface, forming an endocyte, becoming a phagolysosome, and finally digesting within the phagolysosome [23].

As shown in Figure 4A, phagocytosis is driven by the recombination of *F-actin*, which stimulates the diffusion of a pseudopod around the particle and produces phagosomes that are closely apposed to the particle [24]. *F-actin* was up-regulated in the *M. rosenbergii*-infected group at 6 hpi, which might promote the production of phagosomes. C-type lectin receptors (CLRs) play an important role in autoimmunity, allergy, homeostasis, and anti-microbial host defense [25], and one of the functions is to eliminate pathogens by regulating phagosome maturation in macrophages [26]. Up-regulation of the *DCSIGN* gene in the challenge group at 6 hpi might indicate the promotion of phagosome maturation. In addition, V-type proton ATPase can clear the pathogens in the phagosome by increasing the hydrogen ion (H+) concentration [27]. In the present study, the *ATPase* gene was down-regulated at 12 hpi compared to the control group, which might suggest a decreased phagocytic capacity of the phagosome in infected individuals compared to healthy *M. rosenbergii*, and this result is similar to Zhang et al.’s (2019) study which found a significant down-regulation in diseased sea urchins (Strongylocentrotus intermedius) [22].

Crustaceans’ innate immunity depends heavily on lysosomes, which mostly degrade hazardous compounds by phagocytosis and endocytosis, and several lysosomal hydrolytic enzymes, including lipase, proteases, etc., show their greatest enzyme activity at low pH levels [28].

As shown in Figure 4B, LIPA is a very important hydrolase, and reducing its activity can lead to symptoms such as hypercholesterolemia and hepatomegaly [29]. *LYPLA3* is a transacylase that may play a specific role in lysosomes [30]. It is reported that *LYPLA3′*s loss increased atherosclerosis in apolipoprotein E-deficient mice [31]. *ACP2* is the most important lysosomal enzyme in crustacean defense, transferring phosphate groups and catalyzing the hydrolysis of phosphorylated proteins [32]. In addition, loss of *CLN7* will impair mTOR reactivation and the loss of soluble lysosomal proteins [33]. These genes (*LIPA*, *LYPLA3*, *ACP2*, *CLN7*) were up-regulated in the *M. rosenbergii* infection group at 6 hpi, suggesting that they may improve lysosome activity to defend against an *A. veronii* invasion.

### 4.2. Important Immune-Related Genes Involved in the Immune Response

The *Toll* receptor is typically composed of transmembrane regions, intracellular interleukin hormone receptor domains, and extracellular leucine repeats [34]. It is the key vector in immune signal transduction [35]. In the present study, the expression of *Toll* continuously increased in gills for 24 h and remained highly expressed in hemolymph and hepatopancreas at 6 hpi, indicating that the *Toll* gene is involved in the immune response against *A. veronii* infection. In the process of attacking *M. rosenbergii* with viruses (WSSV), Feng et al. found (2016) that the *Toll* gene was uniformly up-regulated within 24–48 h, thus it was speculated that the *Toll* gene played a positive role in the fight against viruses [36]. The result is similar to the present study that the *Toll* expression remained consistently up-regulated in three tissues after injection of *A. veronii*, although it varied across tissues and at different times.

*PAK1* is the first protein kinase gene to be discovered and is extensively expressed in eukaryotic tissues while being crucial for biological processes [37]. In this study, *PAK1* expression was significantly enhanced at 6 hpi in three tissues, then decreased at 12 and 24 hpi successively. The expression level at all time points was higher than the control group. It indicated that *PAK1* was closely related to the immune response in *M. rosenbergii* against *A. veronii*. This result was similar to the study of Ren et al. (2019), who detected *PAK1* in all tissues of infected sea cucumbers (*Apostichopus japonicus*) and the highest expression was found in the coelom. After the silencing of the *PAK1* gene, the lysozyme in the coelom was significantly reduced, thus it was speculated that *PAK1* was involved in the immune response against bacteria [38].

*GSK-3β* plays an important role in innate immunity, phosphorylating and inhibiting glycogen synthase [39]. Ruan et al. (2018) found that down-regulation of *GSK3β* can inhibit WSSV infection, suggesting that it may promote WSSV clearance in *Litopenaeus vannamei* by mediating cell apoptosis [40]. Zhang et al. (2020) also found that *GSK3β* negatively affects *L. vannamei* when it is infected with WSSV by mediating feedback regulation of the NF-κB pathway [41]. Contrary to the above study, *GSK3β* expression in the three tissues of *M. rosenbergii* was remarkably enhanced at 6 hpi in the present study, possibly because these infected individuals were greatly damaged and died in large numbers. The expression of *GSK3β* returned to normal at 12 and 24 hpi, which might be generated by an effective immune response in the surviving shrimp.

IKK (IκB kinase) consists of IKKα, IKKβ, and IKKγ, IKKα, and IKKβ are involved in the phosphorylation of IκB and the activation of NF-κB, while IKKγ is in charge of IKKβ and IKKα’s recruitment [42]. NF-κB is a transcription factor that is essential for the cell cycle, immunity, inflammation, etc. [43,44]. NF-κB proteins are usually inactivated by attaching to the IκB (the κB inhibitor) [45]. IKK will be activated when cells are stimulated from the outside, such as via a virus [46], and the NF-κB protein will be released, leading to inflammatory and immune responses. In this study, the expression of *IKKα* was significantly up-regulated in the hemolymph at 6 hpi and in gills at 12 hpi in the challenge group, suggesting a severe immune response in the prawns at these time points. However, the expression of *IKKα* was down-regulated at 24 hpi, indicating a relief in the symptoms of the surviving prawns. The expression of *IKKα* in the hepatopancreas kept decreasing, suggesting that the hepatopancreas might be an essential immune organ that responds positively to the stimulus from *A. veronii*.

## 5. Conclusions

In this study, a comparative transcriptome analysis revealed a variety of transcriptional information in the hepatopancreas of *M. rosenbergii* at different time points. KEGG enrichment analysis showed that plenty of DEGs were mainly concentrated in the phagosome and lysosome pathways. In addition, the spatio-temporal expression of four immune-related genes was found to be significantly up-regulated at 6 hpi. These findings contribute to understanding the immune mechanism of *M. rosenbergii* infected with *A. veronii* and provide new insights for further research on disease-resistance breeding in *M. rosenbergii*.

## Figures and Tables

**Figure 1 genes-14-01383-f001:**
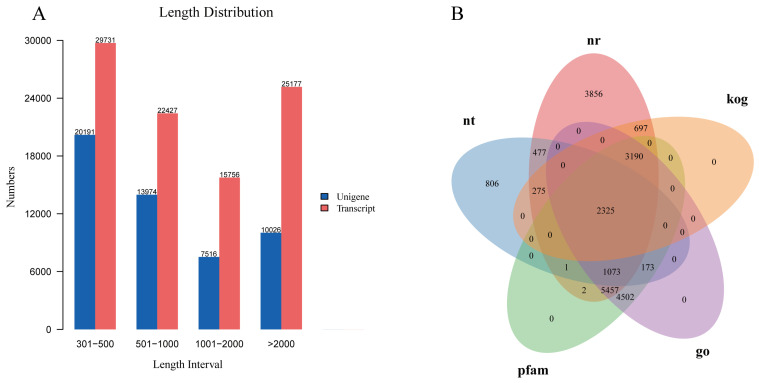
(**A**) The number of unigene (blue) and transcript (red) in different length intervals. The length of unigene/transcript (*x*-axis), and the number of unigene and transcript (*y*-axis). (**B**) Venn diagrams for five databases; each region was annotated with a corresponding number of genes.

**Figure 2 genes-14-01383-f002:**
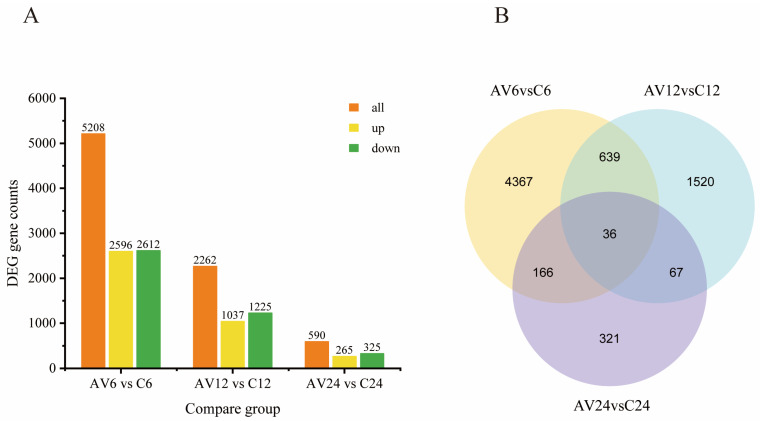
(**A**) Columnar distribution of DEGs in three compare groups. Orange color (all DEGs numbers), yellow color (up-regulation gene numbers) and green color (down-regulation gene numbers). (**B**) Venn diagram reflects the number of DEGs in the three comparison groups.

**Figure 3 genes-14-01383-f003:**
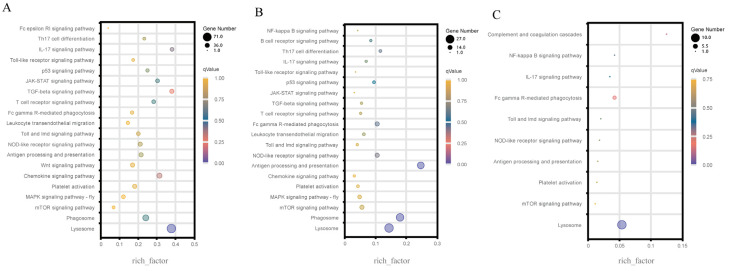
A total of 173 immune-related differential genes (DEGs) were enriched in the top 20 immune-related pathways in the three comparison groups. (**A**) AV6 vs. C6, (**B**) AV12 vs. C12, (**C**) AV24 vs. C24. Dot size represents the corresponding number of DEGs, and color represents enrichment *p*-value.

**Figure 4 genes-14-01383-f004:**
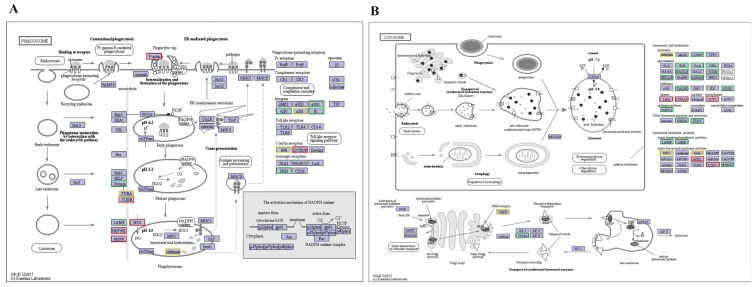
Pathway map of the phagosome (**A**) and lysosome (**B**) pathways in KEGG at 6 h post injection. The red and green boxes represent up-regulated and down-regulated genes, respectively. The yellow boxes represent unclear up-regulation and down-regulation of genes.

**Figure 5 genes-14-01383-f005:**
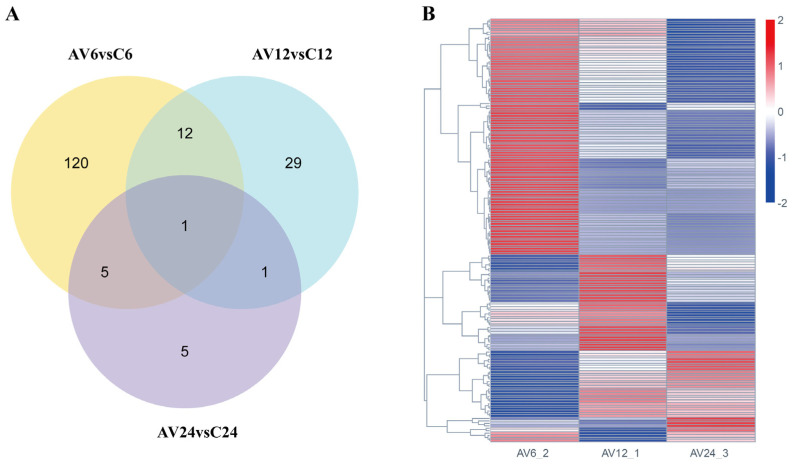
(**A**) Venn diagram shows the number of 173 immune-related DEGs in the three comparison groups (AV6 vs. C6, AV12 vs. C12 and AV24 vs. C24). (**B**) Hierarchical cluster analysis of 173 immune-related DEGs. Red represents high expression and blue represents low expression.

**Figure 6 genes-14-01383-f006:**
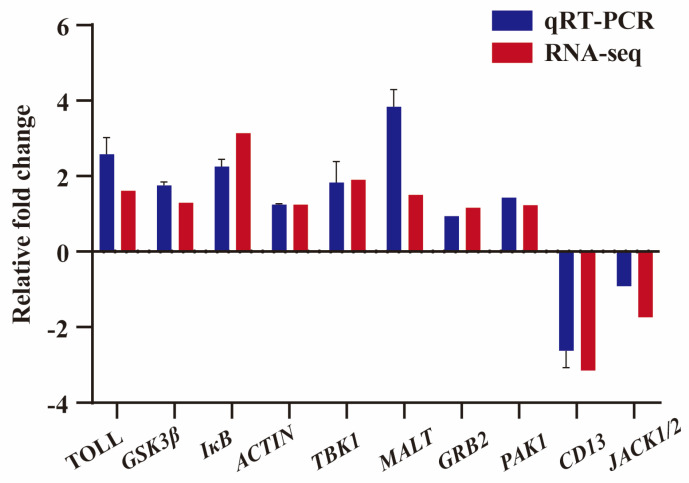
A total of 10 immune-related genes were randomly selected for qRT-PCR validation. The blue column represents the expression level of qRT-PCR, and the red column represents the expression level of RNA-seq. Data shown are the mean of triplicates ± SD.

**Figure 7 genes-14-01383-f007:**
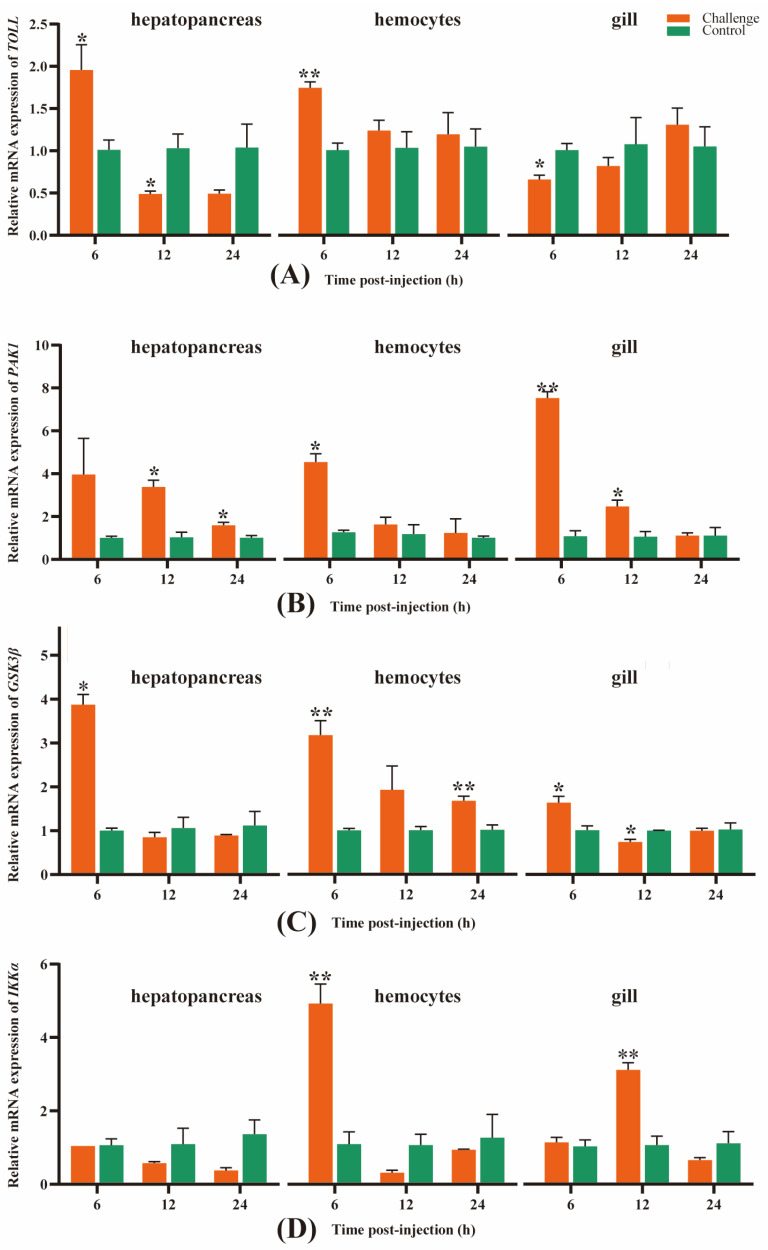
Gene expression analysis of the *TOLL* (**A**), *PAK1* (**B**), *GSK3β* (**C**), and *IKKα* (**D**) in *M. rosenbergii* hepatopancreas, hemolymph, and gills at 6, 12, 24 hpi, respectively. Data shown are the mean of triplicates ± SD. The asterisk represents significant differences between the challenge and control groups at different times (“*”indicates *p* < 0.05 and “**” indicates *p* < 0.01). The orange column and the green column represent the challenge group and the control group, respectively.

**Table 1 genes-14-01383-t001:** Sequences of primers used in this study.

Primer	Gene Name or Usage	Sequence (5′→3′)	Annealing Temperature (°C)
18S-F	Used as reference gene	TATACGCTAGTGGAGCTGGAA	59
18S-R		GGGGAGGTAGTGACGAAAAAT	
GSK3β-F	*Glycogen synthase kinase 3 β*	ACCCGTGAGCAGATTAGA	59
GSK3β-R		GCCTGAAGTGGCGTGATA	
1κB-F	*NF-kappa-B inhibitor α*	GCATAATGGCTATTGAACTG	59
1κB-R		TCCCAAGATGGAACGCTA	
PAK1-F	*p21-activated kinase 1*	TTCGTCGGAAGGTAGAGG	59
PAK1-R		GAGGCTGGTCGGTGGTAT	
TBK1-F	*Tank-binding kinase 1*	AGAGGAGCAAGAAGGTCG	59
TBK1-R		CAGGCTTCAAGTCACGATGT	
TOLL-F	*Protein toll*	CAAACCGTCGGAGGAACA	59
TOLL-R		CCTTGACTGCCACTGAAC	
CD13-F	*Aminopeptidase N*	GAGTGCCGACTTCCAACC	59
CD13-R		CAAGACCTCCAGAACAATA	
Actin-F	*Actin β/γ 1*	ATGGTCGGTATGGGTCAGA	59
Actin-R		AGGTGCTACACGGAGTTCA	
GRB2-F	*Growth factor receptor-Bound protein 2*	GAAGGACTTATTCCCAGCAA	59
GRB2-R		ACCATCGCCACATTTAGG	
IKKα-F	*Inhibitor of nuclear factor kappa-B kinase subunit α*	AATATCCCACTTGAAGCC	59
IKKα-R		CGTTGAAACAGGACGAAA	
MALT-F	*Mucosa-associated lymphoid tissue lymphoma translocation protein 1*	CGGAAGGACGGCGTTACAT	59
MALT-R		CACGGTCACGGGTCTGGTT	
JAK1/2-F	*Janus kinase 1*	AAAGAGCGGATGAGCAC	59
JAK1/2-R		CTGGCAAGTCCCGATGA	

**Table 2 genes-14-01383-t002:** Information of four immune key genes in the immune-related KEGG pathway.

Genes	Fold Change	Up/Down Regulation	Pathway
*Toll*	1.6164	Up	Toll and Imd signaling pathway
*PAK1*	1.2329	Up	Chemokine signaling pathway
			Fc γ R-mediated phagocytosis
			Natural killer cell mediated cytotoxicity
*GSK3B*	1.2959	Up	mTOR signaling pathway
			Chemokine signaling pathway
			Wnt signaling pathway
			T cell receptor signaling pathway
			IL-17 signaling pathway
			B cell receptor signaling pathway
*IKKα*	−5.0217	Down	mTOR signaling pathway
			Chemokine signaling pathway
			NOD-like receptor signaling pathway
			T cell receptor signaling pathway
			Toll-like receptor signaling pathway
			IL-17 signaling pathway
			Th17 cell differentiation
			B cell receptor signaling pathway
			NF-kappa B signaling pathway
			RIG-I-like receptor signaling pathway
			Cytosolic DNA-sensing pathway
			Th1 and Th2 cell differentiation

Note: Gene information in KEGG pathway of *M. rosenbergii* infecting with *A. veronii* at 6 h. In fold change, positive numbers represent gene up-regulation and negative numbers represent gene down-regulation.

**Table 3 genes-14-01383-t003:** Relevant DEGs information in the phagosome and lysosome pathways.

Pathway	Genes	Fold Change	Up/Down Regulation	Genes	Fold Change	Up/Down Regulation
Phagosome	*F-actin*	1.7692	Up	Dvnein	−1.2674	Down
	*TUBB*	2.9491	Up	LAMP	−7.1505	Down
	*PIKFYVE*	1.1451	Up	αVβ5	−3.1955	Down
	*M6PR*	1.315	Up	α2β1	−2.9904	Down
	*NOS*	5.1266	Up	β1	−1.4217	Down
	*DCSIGN*	2.472	Up	SRB1	−2.3453	Down
Lysosome	*LIPA*	1.7805	Up	LGMN	−1.3984	Down
	*LYPLA*	1.6242	Up	GLB	−2.5314	Down
	*ACP2*	1.7805	Up	GAA	−1.8661	Down
	*CLN7*	2.5625	Up	GBA	−1.9262	Down
	*LAMAN*	−5.2917	Down	NAGLU	−2.5669	Down
	*GANLS*	−2.0659	Down	GUSB	−2.8311	Down
	*SGSH*	−2.5976	Down	HXA/P	−3.5577	Down
	*GM2A*	−2.4086	Down	LAMP	−7.1505	Down
	*CLN1*	−1.498	Down	HGSNAT	−1.2058	Down
	*GGAS*	−1.2001	Down			

Note: DEGs information in phagosome and lysosome pathways of *M. rosenbergii* infected with *A. veronii* at 6 h. In fold change, positive numbers represent gene up-regulation and negative numbers represent gene down-regulation.

## Data Availability

All supporting data are included within the main article and all supporting data are included within the main article and its Appendix A.

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
