# Peer review of "The Dynamics of Gene Expression Unraveling the Immune Response of Macrobrachium rosenbergii Infected by Aeromonas veronii"

_genes, 2023, doi:10.3390/genes14071383_

Round 1

Reviewer 1 Report

After reviewing the manuscript titled "The dynamics of gene expression unraveling the immune response of Macrobrachium rosenbergii infected by Aeromonas veronii," the authors investigated the immune response of M. rosenbergii to A. veronii. They conducted comparative transcriptomic analyses of the M. rosenbergii hepatopancreas on both the challenge and control groups at 6, 12, and 24 hours post-infection (hpi). This study was done quite well and the research logic is correct. I believe it will be helpful to this field. I have a few concerns that prevent me from recommending the publication of the paper in its current form. Some revisions are needed to address certain mistakes. Therefore, this manuscript does not currently meet the criteria for publication and requires minor revisions.

Minor comments

1.     L40-L45, In the Introduction section, we suggest the author to emphasize more on the description of the pathogenicity of Aeromonas veronii to help readers understand the severity of this pathogen.

2.     L55-L60, We suggest the author to provide a more precise explanation on the research objective, specifying the problem that needs to be addressed.

3.     L68-L69, The author must explain why the water temperature needs to be set at 28 degrees in the research method.

4.     L131, The information provided in Table 1 is insufficient, and the author should explain the meaning of each primer used.

5.     L134-L138, The author must explain why these four genes were chosen.

6.     L203-L210, The resolution of Figure 3 and Figure 4 is too low, and the author must improve the resolution of these two figures to help readers understand these signaling pathways.

7.     L308-L309, “Crustaceans’ innate immunity depends heavily on lysosomes, which mostly degrade hazardous compounds by phagocytosis and endocytosis.”. There is a need to add references here to support the argument.

The author should add the reference.

8.     L365-L375, In the Conclusions section, we suggest that the authors re-write these sentences based on the 'hypothesis and test' structure—clear questions raised in the introduction, then based on the question, to collect data and to do relevant analyses rather than every analysis, then in the discussion part, go back to the question defined in the introduction. The author's discussion in the abstract and conclusion is too similar. We recommend the author to revise the conclusion to enable readers to clearly understand the important findings and results of this research.

Minor editing of English language required

Reviewer 2 Report

The Manuscript entitled “The dynamics of gene expression unraveling the immune response of Macrobrachium rosenbergii infected by Aeromonas veronii” is an intresting work done by the authors to unravel the immune response of M. rosenbergii after infected with A. veronii.

There are some minor queries needs to be addressed by the authors.

Query 1: First of all the at the discussion part, section 4.1 : phagosome and lysosome pathway analysis. In this section the authors discussed a vast area of immune genes and the function but I am wondering why author did not written anything about this section previously. Again, as it is evident from the section headline that this is pathway analysis, but how did the authors found the result that the DCSIGN gene in the challenge group is upregulated at 6hpi and ATPase gene down regulated at 12hpi.

Also, in figure 4 all these things have been shown but the picture is not that clear to analyze, but the picture did not give any idea about the post challenge pathway or it’s a generalized pathway?

Line 64: and from the sentence must be removed.

Line 73: The author stated that they have done random inspection of the prawn to know that the prawn was free from bacteria. Please describe which type of inspection author has been carried out to see the prawns were free from any bacterial infection.

Line 76: “was first inoculated the tryptic soybean” must be changed to “was first inoculated in the tryptic soybean”

Line 83: The authors have written that they were stored tissue temporarily in liquid nitrogen and kept at -80℃, please recheck it again whether the author snap froze the tissue in liquid nitrogen or kept in liquid nitrogen.

Line 94: Please check the sample named ADV_3.

Line 118: Authors have seen and written about 10-immune related DEGs but in the table 1 there were 11 immune related genes have been shown. Please let us know why this discrepancy.

Line 118-130: Authors did not showcase properly which tissue they have used for validation and how many biological replicates they have used.

Line 145: Please ensure to name the three tissues as (hemolymph, hepatopancreas and gills)

Line 235-236: The authors written that they have seen the gene expression in different tissues, but the author must check whether hemocytes were the tissue or cell.

Line 254: PAK1 gene was remarkably up-regulated expressed…. Please check the sentence.

Reviewer 3 Report

Genes-2397220 review

The dynamics of gene expression unraveling the immune response of Macrobrachium rosenbergii infected by Aeromonas veronii.

In the current work, the author’s sought to evaluate the immune response of M. rosenbergii after exposure to A. veronii using RNA sequencing technology. I find their conclusions interesting; I do however have some comments/suggestions that I hope will be addressed prior to publication.

1.      Figure 4, perhaps it would be better to have a table indicating the DEGs in each pathway and their fold change.

2.      Figure 5, do the different immune pathways correlate with your up and down regulated DEGs. For example, in the phagosome pathway F-actin, Tubb are upregulated and SRb1 and α2β1 are down regulated.

3.      Figure 6, which time point was used for the qPCR analyses.

4.      Minor revision
